# Tribological and Heat Transfer Investigation of Graphene Oxide Coatings on Nylon Rotating Bands in an Artillery System

**DOI:** 10.3390/nano14231943

**Published:** 2024-12-03

**Authors:** Hongbin Chen, Zeyang Meng, Shuang Yi

**Affiliations:** School of Mechanical Engineering, Nanjing University of Science and Technology, Nanjing 210094, China

**Keywords:** artillery systems, rotating band, graphene oxide, coefficient of friction, thermal effect

## Abstract

Exploring ways to improve the performance of rotating bands is of great importance for enhancing the power of modern artillery. This study prepared graphene oxide-coated Nylon (GO-Nylon) and Nylon samples based on nylon rotating bands in artillery systems to investigate the feasibility of introducing GO-coated nylon rotating band materials to enhance their tribological and thermal properties. The friction behavior and thermal effects of these two surfaces were analyzed under different external loads and surface roughness conditions. The results show that the excellent thermal conductivity of GO effectively reduced temperature accumulation during friction. Under an external load of 8 N, the surface temperature of GO-Nylon decreased by 14% compared to the Nylon surface, and the coefficient of friction (COF) decreased by 21%. At the same time, a simulation model was established, and its calculation results were consistent with the experimental trends, providing a further explanation of the experimental phenomena. This research provides a basis for the application of graphene-based coatings in the defense industry and presents new ideas for the development of high-performance rotating band materials.

## 1. Introduction

With the continuous development of the weaponry industry, the demands on rotating band materials for modern artillery have steadily increased. Traditional metal rotating band materials in artillery systems are no longer sufficient to enhance performance. New non-metallic materials have gradually caught engineers’ attention due to their immense potential and have been applied to various components, such as projectiles and barrels, to improve overall artillery performance [1,2]. The rotating band in an artillery system is a crucial link between the artillery and the projectile. Nowadays, nylon often replaces metal as the rotating band material. Based on its primary function, nylon rotating bands can be classified into three types: fixed rotating bands, sliding rotating bands, and gas seal bands. Each type requires different performance characteristics depending on its usage. Despite the functional differences, all types of nylon rotating bands experience strong interactions with the artillery barrel during firing, leading to severe surface wear, high temperatures, and material failure, which significantly affects the functionality of the rotating band.

Many scholars have conducted studies on the behavior of nylon rotating bands during their movement within the artillery barrel. For example, Yang Ming et al. [3] designed a short barrel to study the engraving process of a nylon rotating band under firing conditions, analyzing the macroscopic morphology of the rotating band surface after recovering the fired projectile and improving a three-dimensional dynamic engraving model based on experiments. Wu Bin et al. [4] used an Electronic Universal Testing Machine and a self-made dynamic impact testing device to investigate the static and dynamic characteristics of different materials during the engraving process. Their results showed that the strain rate and temperature significantly influenced morphological changes in the rotating band during engraving. Kailash Kumar et al. [5] studied the dynamic properties of nylon 66, corroborating the findings from [4] experiments and indicating that the strain rate of nylon material has a substantial impact on its properties. In addition to the deformation and failure caused by the interaction between the rotating band and the barrel wall during their movement within the artillery barrel, the friction between them is also a critical factor affecting projectile motion. The friction mechanism of nylon materials differs from that of traditional metals, and their manufacturing processes vary as well. Regarding nylon materials, Zhang et al. [6] employed a novel 3D printing method to fabricate nylon polymer gears and studied their macro- and micro-wear characteristics under traditional and 3D printing processes using a self-made friction wear testing platform. Their findings demonstrated that different manufacturing techniques significantly affect the engineering properties of the materials. In traditional tribological research, employing new shaping techniques during processing and adding friction modifiers to the interacting surfaces can effectively enhance the friction characteristics during material interaction [7,8,9]. For nylon materials, doping with other materials to improve performance has become a common choice based on specific usage requirements [10,11,12]. Carbon-based substances are widely used as dopants due to their excellent physicochemical properties. R. Rafiq et al. [13] found that adding functionalized graphene (FG) to nylon-12 significantly improved its tensile strength, elongation at break, impact strength, and toughness, indicating that the incorporation of functionalized graphene can greatly enhance the mechanical properties of nylon-12.

Graphene and its related materials with its hexagonal honeycomb lattice structure, formed by carbon atom hybridization, have attracted significant attention across various fields due to their remarkable electrical, thermal, and mechanical properties [14,15]. As a surface coating on mechanical structures, graphene can effectively reduce friction and wear between components at both the micro–nano- and macro-scales [16,17,18]. It acts as a friction modifier, promoting macroscopic superlubricity phenomena [19,20]. Due to graphene’s excellent thermal conductivity [21], introducing graphene materials as a cutting fluid during the machining of hard materials can effectively reduce tool wear, cutting forces, and mechanical vibrations during the process [22]. Due to its unique properties, many researchers have prepared graphene composites from different raw materials [23,24] and introduced them to improve friction performance. Haseebuddin, M. R. et al. [25] synthesized graphene-reinforced aluminum-based composites via powder metallurgy and investigated the effect of graphene under different external conditions on the friction and wear resistance of the material through pin-on-disk wear tests. Si-tian Chen et al. [26] doped graphene into iron-based friction materials and studied the effect of graphene doping on friction properties by controlling its content, concluding that the wear rate and friction coefficient decrease with increasing graphene content, and they provided an explanation of the mechanism. Zhan, L. et al. [27] investigated the friction characteristics of graphene at the microscopic scale using atomic force microscopy to study the interaction between graphene and microspherical probes in a microenvironment, explaining the changes in graphene’s friction characteristics under these conditions. Graphene is often combined with different types of substrates to enhance friction performance. D.-H. Cho et al. [28] explored the friction properties of graphene on various materials and its adhesion effects with substrates, revealing the relationship between the bonding strength of graphene with different substrates and its friction performance. Nidhal et al. [29] studied the enhancement in friction welding performance by graphene-reinforced AA6061 material through friction welding (CDFW) experiments, showing that graphene improved the distribution of heat during friction welding, which better balanced the heat generation at the welding site, enhancing the sample’s performance at the welded area.

In the meantime, thermal phenomena are a significant characteristic of friction between objects and a real-world issue for materials like rotating banding during operation. Factors such as external load, surface contact materials, and environmental conditions [30] affect thermal behavior. With the advancement of modern thermal imaging technology, infrared thermography now allows for the observation of heat distribution on interacting surfaces, making it a common technique for studying friction-induced heating in different environments [31,32,33]. Filippov, A. et al. [34] combined an infrared thermal imaging camera with tribology equipment to observe the temperature of wear surfaces of three metals with different thermal conductivities, studying their thermal stability under sliding friction. Ottani, F. et al. [35] proposed a new method of measuring the friction coefficient based on thermal imaging and an inverse heat model. By measuring the temperature changes on the pin surface and combining them with heat generation and conduction properties, they indirectly calculated the friction coefficient, with an error margin of about 12% compared to traditional methods. You, T. et al. [36] used an infrared thermal imaging system based on a charge-coupled device to measure the radiation emitted by the contact area of a sample. After appropriate calibration, they converted the radiation values into temperature, enabling the quantification of thermal degradation during machine operation to identify failures before they occur. Studying the thermal phenomena of materials like nylon and graphene during friction through infrared thermography allows for a better understanding of the changes that occur during the friction process.

This study combines the actual working conditions of existing nylon rotating band materials with the excellent properties of graphene oxide (GO) coatings, aiming to enhance the performance of current rotating bands by incorporating graphene materials. This approach seeks to meet the varying performance requirements of rotating bands in different environments. By employing established tribological testing methods and infrared thermal imaging technology, this study analyzes the friction characteristics and thermal phenomena generated during friction processes. Additionally, it investigates the mechanisms of action based on surface morphology. The findings aim to provide insights for further expanding the application of graphene materials in the weapon industry and for subsequent research on high-performance rotating band materials.

## 2. Materials and Methods

### 2.1. Sample Preparation

Nylon and 20CrNi3MoV are commonly used as materials for modern artillery barrels and rotating bands due to their excellent properties and machinability. In the experiments, these two materials served as the upper and lower samples for friction testing. The upper sample was 20CrNi3MoV, which was provided by Shanghai Li jia Special Steel Co., Ltd. in China, while the lower sample was nylon, supplied by Shenzhen Yihang Plastic Materials Co., Ltd. in China. The TR200 surface roughness measuring instrument of Dongwan Gaotai Testing Instrument Co., Ltd. in China was used to detect the surface roughness of the specimens after processing. To minimize the influence of processing methods on the experimental results, the sample preparation simulated the actual machining processes of the barrel inner wall and rotating band as closely as possible. For the upper sample, a turning tool was used to machine a 10 mm diameter rod of 20CrNi3MoV, creating a semicircular friction contact surface with a radius of 3 mm. For the Nylon lower sample, a combination of wire cutting and turning methods was employed to obtain test pieces with surface roughness values of Ra = 3.2 (μm) and Ra = 6.3 (μm) while maintaining consistent material properties. Graphene nanosheets (ES900561-500MG) sourced from Hong Kong JiSiEnBei International Trade Co., Ltd. in China. After the thermal oxidation method, grephene oxide nanosheets were be obtained. Then, an oxidized graphene–epoxy resin coating was prepared on the surface of the Nylon sample with the following process: First, the Nylon sample was cleaned with water and then subjected to ultrasonic cleaning with anhydrous ethanol for 2 h, after which it was dried in an oven. For the coating preparation, an appropriate amount of oxidized graphene was added to the epoxy resin solution and sonicated for 1 h. This was followed by stirring at 40 °C for 2 h and setting aside. The coating was applied by immersing one surface of the sample in the oxidized graphene–epoxy resin solution for 60 s at room temperature. After removing it, the sample was dried at room temperature and then placed in a constant temperature drying tray to be dried at 40 °C for 2 h, resulting in the final oxidized graphene–epoxy resin coating. The thickness of the coating was approximately 11 μm.

### 2.2. Friction Test

Ball-on-disk tests were conducted by using the UMT-5 (TriboLab) tribometer from Berlin Bruker Nano Inc. in Germany. In the experiment, the upper sample was a 20CrNi3MoV steel hemisphere with a radius of 3 mm, while the lower sample was a Nylon disk with a diameter of 50 mm and a thickness of 4 mm, coated with oxidized graphene–epoxy resin. The relative motion speed between the upper and lower samples was 180 mm/s. The upper sample was subjected to external normal loads, which were 4 N, 8 N, 12 N, and 15 N. Thus, the average contact stresses based on the normal load between upper and lower samples were 66.1 MPa, 83.3 MPa, 95.3 MPa, and 102.7 MPa, respectively. To ensure comparability, two sets of untreated Nylon samples with the same roughness served as control groups. During the tests, an infrared thermal imaging camera monitored the temperature changes on the sample surfaces to analyze the variations in frictional heat phenomena under different contact stresses and surface materials. The monitoring period included the heating phase after friction began until thermal equilibrium was reached at the end of the test. A FLIR A655sc high-resolution thermal imaging camera was used, featuring a maximum frame rate of 50 Hz and a temperature measurement range of −40 °C to 150 °C, adequately meeting the requirements for temperature measurements during the experiment. The thermal imaging lens was positioned 20 cm above and angled towards the rotating friction area of the lower sample to capture the surface temperature changes during friction. During the experiment, in order to observe the thermal phenomena during the friction process while obtaining a stable friction coefficient, 60 s was selected as the test time. This section’s experiments were conducted three times for each group, and the results represent the average of the three trials.

### 2.3. Simulation Design

In this study, finite element analysis was employed to simulate the thermal phenomena occurring during the ball-on-disk friction tests for better understanding. After establishing CAD models for the upper sample (20CrNi3MoV steel hemisphere) and the lower sample (Nylon disk), a C3D8RT mesh model was created in ABAQUS 2021, incorporating external conditions for calculations. The length, width, and height of the mesh at the contact area were 0.404 mm, 0.0625 mm, and 0.250 mm, respectively. The upper sample was allowed to move only in the vertical direction, while the lower sample was permitted to rotate around the vertical axis. The simulation used a dynamic explicit thermal analysis step to compute the heat generation process due to friction, focusing on the thermal phenomena produced under different external loads and rotational speeds. Given that the hardness of the 20CrNi3MoV steel hemisphere far exceeded that of the Nylon disk, the upper sample was simplified as a rigid body in the calculations to enhance computational efficiency.

## 3. Results and Discussion

### 3.1. Tribological Performance

The macro-friction performance of the GO-coated Nylon (GO-Nylon) and Nylon surfaces was investigated with different surface roughness and external normal load conditions while interacting with 20CrNi3MoV steel at a relative motion speed of 180 mm/s. The changes in the coefficient of friction (COF) for both surface types were analyzed at different surface roughness levels (Ra = 3.2, Ra = 6.3) and four external normal loads (4 N, 8 N, 12 N, 15 N) (Figure 1a–c). Figure 1a illustrates the variation in the COF of the Nylon and graphene oxide-coated Nylon (GO-Nylon) surfaces over time. It can be observed that the GO-Nylon surface underwent a distinct break-in period, during which the COF value fluctuated significantly between 0 and 20 s, rapidly increasing from a minimum value of 0.199 to 0.321. This indicates that during the initial contact phase, there may be a process of lubricant coating or structural adjustment on the GO-Nylon surface. Following this period, the COF stabilizes and remains at around 0.32. In contrast, the COF of the Nylon surface is relatively steady from the beginning, starting at 0.34 and gradually increasing, eventually stabilizing at approximately 0.394 by 50 s. GO-Nylon demonstrates a lower stable COF compared to Nylon, with a reduction of around 18.7%. Figure 1b shows a comparison of the COF at different surface roughness levels (Ra = 3.2 and 6.3). Under roughness levels of Ra = 3.2 and Ra = 6.3, the average COF of the GO-Nylon surface showed a reduction compared to the original Nylon surface. However, this change was not very pronounced between the different surface roughness levels, indicating that surface roughness had minimal impact on the COF of the two materials. For the effect of normal load, Figure 1c illustrates the variation trend in the COF under different normal loads (4 N, 8 N, 12 N, 15 N) for both Nylon and GO-Nylon. As the external load increased from 4 N to 15 N, the COF curve for Nylon exhibited fluctuating changes. In contrast, GO-Nylon showed only a slight increase in the COF during this process. At a low external load of 4 N, the COF values for the GO-Nylon and Nylon surfaces were 0.28 and 0.22, respectively. As the external load increased to 8N, the average COF of the GO-Nylon surface decreased by 21% compared to the Nylon surface. However, at higher external loads such as 12 N and 15 N, this improvement effect was noticeably reduced.

In the meantime, the friction force, as the ultimate result of friction process improvements (Figure 1d–f), displayed trends similar to the COF values. Figure 1d shows the variation in the friction force over time for the Nylon and GO-Nylon surfaces at Ra = 6.3. The friction force of Nylon increased rapidly during the first 10 s and then gradually stabilized. In contrast, GO-Nylon exhibited significant fluctuations in the friction force, starting from 2.83 N, gradually increasing to 5.24 N, and stabilizing after 30 s, followed by a gradual decrease. Additionally, the friction force under different surface roughness conditions (Ra = 3.2 and Ra = 6.3) is shown in Figure 1e. It can be observed that for Ra = 3.2, the friction force stabilized at around 2.62 N after 50 s. In contrast, the friction force increased significantly and reached 2.73 N at 40 s for Ra = 6.3, indicating that higher surface roughness (Ra = 6.3) increases the friction force. Figure 1f shows the variation in the friction force over time under different loads (4 N and 12 N). At 4 N, the friction process on the GO-coated surface did not exhibit a clear break-in period, and the friction force variation curves for both surfaces were very similar. At 12 N, the running-in process was more evident, with the friction force stabilizing after 40 s.

### 3.2. Surface Morphology Analysis

The surface morphology and friction marks of both the upper and lower samples after friction were observed using a 3D white light interferometer and an optical microscope (Figure 2a–h). Under the conditions of surface roughness Ra3.2 and an external load of 12 N, the surface of the upper sample, a 20CrNi3MoV steel hemisphere, was examined with a 3D white light interferometer (Figure 2a). The surface profile showed an inclined plane with raised ridges, and a significant indentation (2 μm) was observed at the center. In comparison, the steel ball interacting with the GO surface under the same conditions showed different characteristics (Figure 2b), where the surface profile at the same white line position as in Figure 2a displayed no noticeable indentation on the inclined plane. The lower sample provided a contrast to the upper one. In Figure 2c, under the same conditions of surface roughness Ra 3.2 and an external load of 12 N, the Nylon surface of the lower sample disk exhibited distinct wear marks at the location indicated by the white dashed line, with a depth of 9.29 μm. Under the same external conditions, the wear depth on the GO-coated Nylon disk (Figure 2d) exceeded that of the original Nylon surface, reaching 17.10 μm. Comparing the undamaged regions of both surfaces revealed that the profile of the GO-Nylon disk was sharper than that of the original Nylon surface by about 13 μm, indicating that this difference was due to the GO coating on the surface of the sample.

Figure 2e–h show the optical microscope images of the wear areas on the surfaces with different roughness (Ra = 3.2 and 6.3) and under the same external load (12 N). Notably, wear debris remained on the upper sample surface, which likely originated from the transfer of GO material from the lower sample surface during friction. At surface roughness Ra = 6.3, under 100× magnification (Figure 2(e1)), black residue and large amounts of strip-shaped wear debris were observed. Under 1000× magnification (Figure 2(e2)), the shape of the wear debris and nearby black material became clearer. The corresponding lower sample (Figure 2f) showed a wear scar width of 945 μm (Figure 2(f1)), and under 400× magnification, residual wear debris on the friction surface was visible (Figure 2(f2)), displaying a similar shape to the debris seen in Figure 2(e2). At surface roughness Ra = 3.2, the wear debris on the upper sample significantly decreased under 100× magnification (Figure 2(g1)). Unlike in Figure 2(e1), the black marks were left on both ends of the upper sample’s spherical surface in a defined direction. Under 1000× magnification (Figure 2(g2)), black material similar to that seen in Figure 2(e2) was observed, with its distribution corresponding to the direction of friction. The corresponding lower sample (Figure 2h) showed a reduced wear scar width of 654 μm at surface roughness Ra = 3.2 (Figure 2(h1)). Observing the wear area under 400× magnification revealed a noticeable reduction in wear debris (Figure 2(h2)) compared to Figure 2(f2). Based on the changes in wear scar width, it can be concluded that under the same external load (12 N), the degree of wear on the sample surface is positively correlated with surface roughness.

### 3.3. Thermal Imaging Analysis

The infrared thermal imaging camera captured the thermal phenomena on the sample surface during high-speed friction in the wear testing machine. Figure 3a shows the arrangement of the two main instruments used in the experiment. The infrared thermal imaging camera was fixed next to the wear testing machine using a tripod, maintaining a stable position relative to the machine’s tray, which allowed for consistent recording of temperature changes on the lower sample surface during friction. Figure 3b displays the trend of surface temperature changes over friction time for the Nylon and GO-Nylon samples under different loads (4 N, 8 N, 12 N, 15 N). Under a 15 N load, the surface temperature of Nylon gradually increases from room temperature (approximately 25 °C) and reaches a peak (about 45.3 °C) at around 60 s. Under 8 N and 4 N loads, the temperatures rise to approximately 51.14 °C and 37.27 °C, respectively. Compared to Nylon, the temperature rise of GO-Nylon is significantly reduced. Under an 8 N load, the maximum temperature for GO-Nylon is about 51 °C, approximately 8 °C lower than that of Nylon, demonstrating the effective heat suppression provided by GO. At a 4 N load, the temperature of GO-Nylon remains around 37.27 °C, which is higher than the 36.38 °C of regular nylon. Therefore, it could be seen that the temperature follows a typical friction-induced heating curve, where it gradually increases to a peak and then stabilizes. After friction stops (around 60 s), the surface temperature of the samples begins to drop rapidly, indicating that heat dissipation mainly occurs through surface cooling. The cooling rate of GO-Nylon is faster compared to Nylon, suggesting that it has a higher thermal conductivity, allowing it to dissipate heat more quickly.

To better understand the thermal distribution on the sample surface during friction, the width of the thermal friction ring captured by the infrared thermal imaging camera was measured at fixed times. Images taken at 1 s and 20 s after the start of the friction test were selected for measurement, capturing a 50 × 50-pixel area at the location of the friction ring, which was then enlarged to 200 × 200 pixels. In each test group, the width of the friction ring was measured four times, and the average value was calculated. Figure 3c presents the measurement results, showing that the width of the thermal friction ring on the GO-Nylon surface was consistently greater than that on the Nylon surface. At 20 s, both surfaces exhibited a similar trend, with the width of the thermal friction ring initially increasing and then decreasing as the external load increased. Figure 3d–f demonstrates the infrared thermal imaging of the heat transfer process for Nylon and GO-Nylon under different conditions. It could be seen that the heat is concentrated in the frictional contact area at the beginning of friction (as seen in the “frictional heat generation region” in Figure 3(d2)), where the temperature rises significantly, forming a high-temperature zone. As friction continues, the heat gradually spreads outward, creating a larger heat transfer region. This wide range of heat diffusion indicates that during the friction process of Nylon, substantial frictional heat accumulates on the surface, leading to concentrated temperature and heat conduction over a broader area. The temperature in the frictional heat generation region is relatively low, indicating that GO helps reduce the accumulation of frictional heat. Compared to Nylon, the heat diffusion region of GO-Nylon is more limited, suggesting that its thermal conductivity is better, allowing heat to dissipate more quickly from the friction interface, thereby preventing excessive heat build-up. For Nylon, the heat dissipation area was significantly reduced, leading to a higher peak surface temperature. In contrast, although the temperature in the heat generation zone of graphene GO-Nylon also increased noticeably, the overall temperature distribution was more uniform, the heat transfer area was larger, and the peak surface temperature showed a significant reduction.

### 3.4. Simulation Analysis

To analyze the thermal accumulation at different rotational speeds during friction, the existing commercial software ABAQUS 2021 was utilized for calculations. Abaqus offers powerful simulation capabilities that effectively analyze the thermal phenomena generated during friction. During the construction of the two friction components, simplifications were made (Figure 4a), with the lower sample disk modeled as a ring to maintain consistent mesh geometry. The mesh was refined at the contact points to ensure calculation quality. The frictional heat was primarily concentrated in the contact area and spreads across the surface through rotational friction. This model was utilized to study the temperature rise and heat diffusion behavior in the contact region under different load conditions during the friction process. Figure 4b displays the calculation results for frictional heat generation under four different normal loads (4 N, 8 N, 12 N, and 15 N), with a focus on the heat rise behavior in the frictional contact area. The temperature exhibits a stepped increase over time, indicating that the heat generation during the friction process accumulates gradually. This stepwise accumulation of heat with each friction cycle leads to a continuous rise in temperature. The increase in frictional heat under high loads indicates that the amount of frictional heat generated is directly proportional to the applied load. After averaging the temperature increases and decreases at fixed points following each contact (Figure 4c), it was evident that the temperature change amplitude at these points increased with the external load, aligning with expected behaviors. Figure 4d–f present finite element simulations of the temperature field distribution during the friction process, illustrating the changes in surface temperature at 0 s, 0.25 s, and 0.5 s. At the beginning of the friction process, the entire surface temperature was uniformly distributed at 26 °C (Figure 4d). After 0.25 s of friction, the temperature in the contact area began to rise. As the process continued, the heat gradually spread outward from the contact area, forming a distinct diffusion zone (Figure 4e) and indicating that the heat was primarily transmitted along the contact interface and then gradually diffused outward. The simulation results have explained the reasons for the changes in thermal phenomena among different control groups under various external environments during the friction process. However, no effective explanations have been provided for other issues during the friction process, such as other complex behaviors of the materials and the impact of the environment on the overall thermal effect during the rotational friction experiment. To address these issues, it will be necessary to couple more environmental and material parameters in combination with subroutines in future simulation processes.

### 3.5. Friction Mechanism

Based on the above analysis, the schematic diagram of the friction, wear, and thermal effects between the 20CrNi3MoV sample and the GO-Nylon surface is shown in Figure 5. At the start of the friction test, the steel ball sample begins the break-in process with the GO-Nylon surface. During this phase, the relative contact area and surface lubrication characteristics continually change, leading to an unstable friction process, especially evident in the break-in stage (Figure 1a). The COF exhibits significant fluctuations initially, but these fluctuations gradually decrease and stabilize over time, with the GO-Nylon surface forming a stable lubrication layer between the original nylon matrix and the 20CrNi3MoV sample.

The formation of this lubrication layer is closely related to the external load magnitude. Four different external loads were tested; at a low load (4 N), no effective lubrication layer was formed. As the external load increased (8 N), the lubrication effect became evident, creating a good graphene lubrication layer in the friction interface (Figure 5(c1)). However, with further increases in load, this lubrication effect diminished, leading to a notable reduction in graphene thickness between the samples (Figure 5(c2)). At the maximum load (15 N), the graphene coating at the center of the wear track exhibited significant damage, resulting in a mixed contact situation between the upper sample, the graphene coating, and the lower Nylon substrate (Figure 5(c3)). During this phase, the graphene provided partial lubrication, while the Nylon surface began its break-in process (Figure 1d).

The thermal phenomena on the GO-Nylon surface during friction correlated with the COF and friction force experimental data. A rapid increase in temperature was observed on the friction disk surface, with a noticeable peak compared to the Nylon surface. This corresponds to the break-in phase of the GO-Nylon surface, which generates substantial heat due to surface degradation. After entering the stable phase, the thermal changes on the GO-Nylon surface leveled off, with temperatures at the same location generally lower than those on the Nylon surface, attributed to graphene’s excellent thermal conductivity (Figure 5d,e). As shown in Figure 3b,c, the heat generated during friction was rapidly dispersed by the graphene surface to nearby areas, whereas the original Nylon surface had a weaker heat dissipation capacity, leading to localized heat accumulation and higher temperatures. The retained two-dimensional planar structure of graphene in graphene oxide is the key factor for the improvement in thermal conductivity. Although oxygen-containing functional groups are introduced, the planar structure of graphene is still preserved in graphene oxide. This structure enables the connections between carbon atoms to be tight and regular, providing a good foundation for the efficient transfer of heat within the plane.

## 4. Conclusions

The objective of this study is to investigate the effect of GO on the tribological and thermal characteristics of nylon driving bands. Tribological tests were conducted on GO-Nylon and Nylon surfaces under varying external loads and surface roughness conditions. Infrared thermography was used to analyze the thermal phenomena during these tests. Based on the experimental results, the following conclusions were drawn:(1)The tribological performance of the GO-Nylon surface has significantly improved compared to the original Nylon surface. Under external loads of 8 N, 12 N, and 15 N, the COF values of the GO-Nylon surface decreased by 21%, 6.25%, and 3.13%, respectively, compared to the original surface. This phenomenon is not as apparent under lower external loads, suggesting that the improvement is closely related to the magnitude of the external load.(2)The results from optical microscopy and 3D white light images show that GO can be uniformly distributed on the Nylon surface. Using the preparation method employed in the experiment, the GO coating maintained a stable distribution of approximately 13 μm on the original Nylon material surface.(3)The thermal effects during the friction process of the GO-Nylon surface were significantly reduced. Under different external loads of 8 N, 12 N, and 15 N, the peak temperature at the frictional contact points decreased by 14%, 5.4%, and 8.16%, respectively. Compared to the Nylon surface, the temperature distribution on the GO-Nylon surface was more uniform, reducing excessive heat concentration on the friction surface.(4)The simulation results aligned with the experimental trends under different external conditions, indicating that the heat generated from a single friction event is positively correlated with the external load.(5)At the present stage, the production technology of graphene oxide materials has matured, and it hardly has any impact on the environment during the production process. Through the improvement in the friction and thermal properties of graphene oxide materials when used in rotating band materials, we believe that it can be applied to other similar working conditions to enhance the basic properties of the original materials.

## Figures and Tables

**Figure 1 nanomaterials-14-01943-f001:**
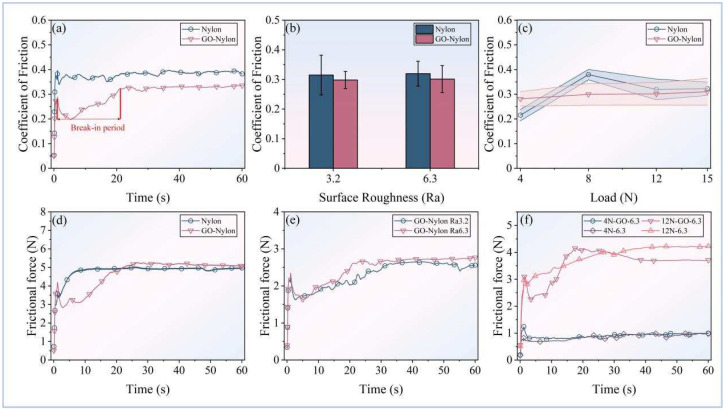
Results of friction experiments under various working conditions. (**a**) Comparison of the coefficient of friction (COF) over time for the GO-Nylon and Nylon surfaces at a surface roughness of Ra = 6.3 and an external load of 12 N. (**b**) Average COF of the GO-Nylon and Nylon surfaces at different roughness levels under an external load of 12 N. (**c**) Average COF of the GO-Nylon and Nylon surfaces at different external loads with a surface roughness of Ra = 6.3. (**d**) Variation in the friction force over time at a surface roughness of Ra = 6.3 and an external load of 15 N. (**e**) Comparison of the friction force over time at different roughness levels under an external load of 8 N. (**f**) Comparison of the friction force over time for the GO-Nylon and Nylon surfaces under external loads of 4 N and 12 N with a surface roughness of Ra = 6.3.

**Figure 2 nanomaterials-14-01943-f002:**
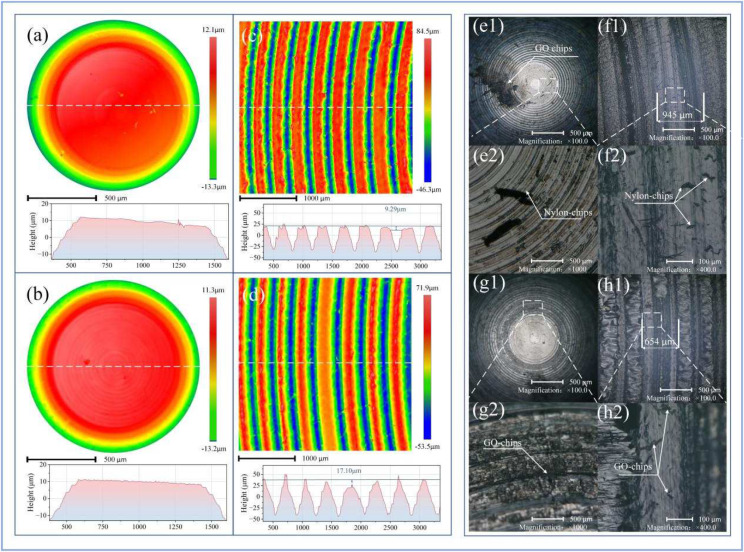
Three-dimensional white light interferometric images and optical microscope observations of the sample surfaces. The white dashed lines indicate the locations of the profile scans. (**a**) White light image of the upper sample surface during friction with the Nylon surface at a surface roughness of Ra = 3.2 and an external load of 12 N. (**b**) White light image of the upper sample surface during friction with the GO-Nylon surface at a surface roughness of Ra = 3.2 and an external load of 12 N. (**c**) Corresponding white light image of the Nylon surface on the lower sample for (**a**). (**d**) Corresponding white light image of the GO-Nylon surface on the lower sample for (**b**). (**e**) Optical microscope image of the wear marks on the upper sample surface at a surface roughness of Ra = 6.3 and an external load of 12 N. (**f**) The observation result under the optical microscope corresponding to (**e**). (**g**) Optical microscope image of the wear marks on the upper sample surface at a surface roughness of Ra = 3.2 and an external load of 12 N. (**h**) The observation result under the optical microscope corresponding to (**g**).

**Figure 3 nanomaterials-14-01943-f003:**
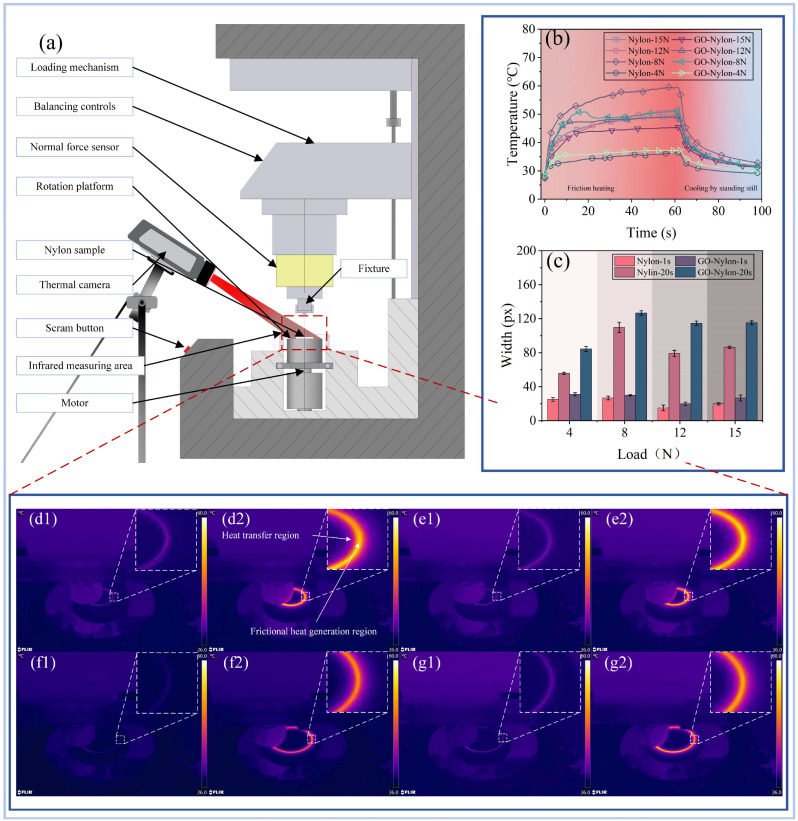
Observational results of the experimental process using thermal imaging equipment. In the thermographic images (**d**–**g**), Part 1 shows the thermal phenomena on the surface when the friction lasts for 1 s, and Part 2 shows the thermal phenomena on the surface 20 s after the friction. (**a**) Overall schematic of the experimental setup, primarily consisting of an infrared thermal imaging camera and a friction wear testing machine. The camera lens is aligned with the rotating platform where the lower sample is located to measure the temperature variations on the surface during friction. (**b**) Changes in surface temperature of the lower sample in the same area under different external loads when the surface roughness is Ra = 6.3. (**c**) Width of the friction ring on the GO-Nylon and Nylon surfaces after rotating for 1 s and 20 s under different external loads with a surface roughness of Ra = 6.3. (**d**) Thermal images of the Nylon specimen under an external load of 8 N after rotating for 1s and 20 s. (**e**) Thermal images of the GO-Nylon specimen under an external load of 8 N after rotating for 1s and 20 s. (**f**) Thermal images of the Nylon specimen under an external load of 15 N after rotating for 1 s and 20 s. (**g**) Thermal images of the GO-Nylon specimen under an external load of 15 N after rotating for 1 s and 20 s.

**Figure 4 nanomaterials-14-01943-f004:**
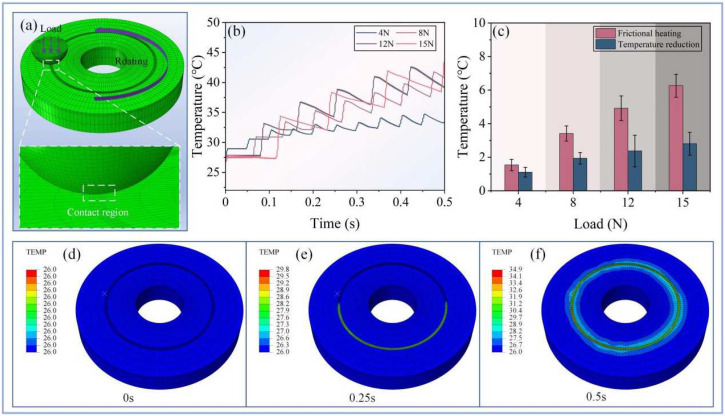
Simulation analysis of thermal phenomena during the friction process on the surface of Nylon. (**a**) Schematic of the finite element mesh during the simulation process. (**b**) Temperature variations on the disk under different external loads with the same relative speed but different rotation speeds. (**c**) Trends in the temperature increase and decrease due to frictional heat generation at fixed points on the upper and lower samples during rotation in the simulated environment. (**d**–**f**) Changes in surface temperature at three time nodes when the external load is 4 N and the rotation speed is 1200 r/min.

**Figure 5 nanomaterials-14-01943-f005:**
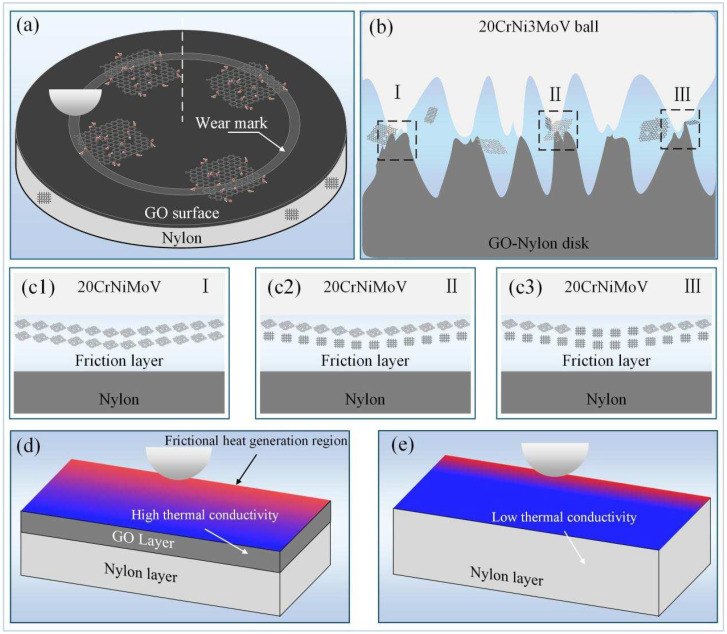
Mechanism of surface friction and thermal effects. (**a**) The friction process between the upper sample and the GO-Nylon surface. (**b**) Contact situation between the upper sample and the lower sample. (**c1**) The state of the friction layer under low external loads. (**c2**) The state of the friction layer under medium external loads. (**c3**) The state of the friction layer under high external loads. (**d**) Thermal dissipation principle of GO-Nylon material. (**e**) Thermal dissipation principle of Nylon material.

## Data Availability

The original contributions presented in this study are included in the article/Appendix A. Further inquiries can be directed to the corresponding author.

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
