# Peer review of "Tribological and Heat Transfer Investigation of Graphene Oxide Coatings on Nylon Rotating Bands in an Artillery System"

_nanomaterials, 2024, doi:10.3390/nano14231943_

Round 1
Reviewer 1 Report
Comments and Suggestions for Authors
In this study, the authors have described an investigation of the tribological behaviour of nylon coated with graphene oxide sliding against steel with that of uncoated nylon. The graphene has the effect of reducing the coefficient of friction, as well as greater thermal dissipation from the contact zone. The authors also used FE modelling to compare the temperatures with those obtained in the experimental measurements, with reasonable agreement between the two.
Although the work described in the manuscript is of interest, it needs some revision before it can be accepted for publication. Some suggested revisions are listed below.
1. The authors do not appear to have stated the thickness of the coating. Although they mention in the Conclusions section that “…the GO coating maintained a stable distribution of approximately 13 μm…” (line 439), this is the first time it is mentioned in the text.
2. In several places in the manuscript, the authors quote surface roughness (Ra) values without units. I know that they are probably in micrometres, but they still need to be given. See, for example, lines 140, 190, 202, 222, 223, 229 and 288 (this list is not exhaustive).
3. Section 2.1 (line 143): at what temperature was it dried in the oven?
4. Section 2.1: the authors state that test pieces of two different Ra values were used in the measurements, but how were the roughness measurements made? The authors discuss white light interferometry of the wear surfaces (section 3.2): was this technique used for the surface roughness measurements as well? Details of the equipment used should be included in Section 2.1.
5. Section 2.2: please state the sliding speed. Although it is given further down the page in Section 3.1, all test conditions should be given in section 2.2.
6. Section 2.2: what were the reasons for choosing the test conditions? How representative were they of actual in-service test conditions? Is this why the ball-on-disc tests were only 60 s, which seems quite short.
7. Section 2.3: what was the mesh size?
8. Section 2.3 (line 175): should this not read “…ABAQUS commercial simulation…”?
9. Section 3.1 (line 203): the authors state that the GO-Nylon specimen shows a “significant reduction” in friction coefficient compared to the original Nylon surface. However, this does not seem to be supported by Figure 1b, in which the differences in COF seem to be much less than the error bars: this suggests that the differences are not that significant after all.
10. Section 3.1 (line 208): for some reason, COF has become “covalent organic framework” rather than “coefficient of friction”. This makes no sense in the present discussion.
11. Figure 1: the caption for Figure 1d mentions the “Ra = 6.3” specimen; however, the discussion of Figure 1d in the text (line 229) mentions “Ra = 3.2”. Which one is correct?
12. Figure 2: the discussion of Figure 2c in the text (lines 252/253) mentions the “Nylon surface”, while the caption for Figure 2c (line 266) states “GO-Nylon”. As with the previous point, which one is correct?
13. Section 3.2 (line 276): here, the authors mention the transfer of GO material from the Nylon to the steel ball. What observations have they made to support this statement? Although they have some optical micrographs, some scanning electron micrographs should help to show this in more detail. They could also use energy dispersive spectroscopy (EDS) to identify the chemical constituents of the transferred material, while Raman spectroscopy could help to show that it is graphene.
14. Figure 4: please add to this caption that it is for uncoated Nylon.
15. Figure 5: Please provide further details in the figure caption for figures 5c1, 5c2 and 5c3. The differences between these three images are not obvious.
Reviewer 2 Report
Comments and Suggestions for Authors
The article deals with a very current topic in the field of application of advanced materials for improving the performance of artillery systems, with a focus on the tribological and thermal characteristics of graphene oxide as a coating. The article contributes to the field of graphene applications in tribological systems. I recommend his publication with minor changes.
-The authors could expand a part on interactions between graphene oxide (GO) and nylon at the molecular level, specifically how the presence of graphene affects friction and heat dissipation. Currently, the results show a reduction in the friction coefficient and an improvement in heat dissipation, but a detailed discussion of the mechanism behind these phenomena is missing. Add explanations based on relevant literature on how the molecular structure of graphene contributes to friction reduction and increase of thermal conductivity.
-Unify size and type of text in whole paper.
-Experimental results have successfully validated the simulations but, it would be useful to include more details about the simulations themselves. How the initial conditions are set in ABAQUS simulations, and are there any restrictions that might affect the accuracy of the results? Include a discussion of numerical model limitations and how they could be overcome in future research.
-Explain why specific values ​​of load (4 N, 8 N, 12 N, 15 N) and surface roughness (Ra = 3.2, Ra = 6.3) were chosen in the experiments. Were these values ​​chosen based on previous research, industry standards, or other factors? Add explanation relating these parameters to actual conditions in artillery systems or similar applications.
-Expand the graphical representations of the results to more clearly see the trends in the data. With addition of more thermal images from infrared thermography heat distribution can be better understood.
-Compare results obtained with results from other researchers.
-Was it possible to measure wear rate? What was the wear scar width?
-Are these materials eligible for mass production or are there any limitations?
-Between number and the unit there should be space, please unify.
-If coatings are used over long periods or under different conditions (e.g., temperature, humidity)how would the wear resistance behave?
-Consider the environmental aspects of the graphene oxide application, especially in the context of industrial applications. A section could be added that considers the sustainability of graphene production and its influence on the environment.
-Expand the conclusion, add the potential applications of developed materials beyond artillery systems.
Round 2
Reviewer 1 Report
Comments and Suggestions for Authors
I would like to thank the authors for revising their manuscript. However, there are a few points I made that they have not addressed to my satisfaction. These are listed below.
The authors have not answered my question (comment 2 in my previous review) about the units used for the surface roughness values they quote in their paper. In the absence of units, these values are meaningless. I accept that they do not need to quote the units every time they are mentioned in the text, but the first time they appear (Section 2.1, line 140) the units should be given. In subsequent places in the text, they can then be referred to as “Ra 3.2” or “Ra 6.3” for brevity. This is not an unreasonable request.
Another point related to the above is my question on how the roughness was measured (comment 4 in my previous review). Again, the authors have not answered my question; instead, they have just described the production of the surfaces by machining, which was not what I asked.
In their response to comment 6 of my previous review, the authors stated the purpose of selecting 60 s as the experimental time should be added to Section 2.2 to explain their reasoning for the test conditions chosen.
Comment 13: please add the Raman spectrum (shown in their response) to the manuscript.
These points listed above must be addressed before this manuscript can be accepted for publication.
Round 3
Reviewer 1 Report
Comments and Suggestions for Authors
The authors have addressed most of my previous comments. Regarding my question (comment 2) about how the roughness was measured, from the image provided by the authors of the equipment used, it looks like a stylus profilometer. The authors should mention the equipment manufacturer and model name/number in Section 2.1.
